# Latent Profiles of Premorbid Adjustment in Schizophrenia and Their Correlation with Measures of Recovery

**DOI:** 10.3390/jcm11133840

**Published:** 2022-07-02

**Authors:** Alejandra Caqueo-Urízar, Felipe Ponce-Correa, Carla Semir-González, Alfonso Urzúa

**Affiliations:** 1Instituto de Alta Investigación, Universidad de Tarapacá, Arica 1000000, Chile; 2Programa Doctorado en Psicología, Escuela de Psicología y Filosofía, Universidad de Tarapacá, Arica 1000000, Chile; fponcec@academicos.uta.cl; 3Escuela de Psicología y Filosofía, Universidad de Tarapacá, Arica 1000000, Chile; carla.semir.gonzalez@alumnos.uta.cl; 4Escuela de Psicología, Universidad Católica del Norte, Antofagasta 1270709, Chile; alurzua@ucn.cl

**Keywords:** premorbid adjustment, schizophrenia, recovery

## Abstract

Premorbid adjustment (PA) has classically been defined as psychosocial functioning in the areas of education, occupation, social and interpersonal relationships prior to evidence of characteristic positive symptomatology. It is a concept which possesses ample evidence regarding its predictive nature for the course of Schizophrenia. The study aimed to analyze the latent profiles of premorbid adjustment and their relationship with symptomatology, functionality, subjective recovery, stigma resistance and years of untreated psychosis. Latent class analysis (LCA) was used to elaborate a solution of three premorbid adjustment profiles in a sample of 217 patients diagnosed with Schizophrenia from Public Mental Health Centers in the city of Arica, Chile. The results show that premorbid adjustment was significantly correlated with recovery indicators and that latent profiles of better premorbid adjustment predict better outcomes in subjective recovery and stigma resistance. The results show that premorbid adjustment not only has implications for the severity of the disorder, but that psychosocial functioning prior to psychosis affects the patient’s subjectivity, the representation of the disorder and the recovery process.

## 1. Introduction

Schizophrenia is considered a severe neuropsychiatric disorder characterized by heterogeneous symptomatology, cognitive impairment and significant behavioral dysfunction [1,2], and affects approximately 23.6 million people worldwide [3,4,5]. Compared to the general population, schizophrenia poses a high impact on people’s quality of life reducing life expectancy by 10 to 25 years and with high rates of premature mortality [6,7]. It is also among the 15 leading causes of disability worldwide, with a prevalence of 0.28% and a heavy associated economic burden, so its understanding continues to represent a public health imperative [8,9,10].

Although schizophrenia has traditionally been described as a chronic disorder with an unfavorable clinical course, in the last decade, research supports the validity of considering recovery in schizophrenia an empirically quantifiable construct [11,12,13]. Recovery is a process that incorporates different perspectives not necessarily concordant with each other, encompassing both the patient’s and the healthcare team’s point of view [14]. It is possible to find clinical components such as symptom remission and functionality [14,15], and subjective components associated with the development of individualized coping mechanisms and improvements in levels of psychological well-being [16,17].

Premorbid adjustment (PA) has classically been defined as psychosocial functioning in the areas of education, occupation, social and interpersonal relationships prior to evidence of characteristic positive symptomatology [18]. Premorbid adaptation in schizophrenia is considered a psychosocial or phenotypic expression of brain alteration that precedes psychosis in apparently healthy adolescents who subsequently develop schizophrenia [19,20]. It is a concept that has ample evidence regarding its predictive nature for the course of Schizophrenia [21,22,23,24].

Research has shown that poor premorbid adjustment predicts worse symptomatology and cognitive impairment [20,25,26], diagnostic severity [21], severity in theory of mind deficits [23], greater self-stigma [27] and low response to treatment [28,29,30,31] Conversely, better premorbid adjustment scores are associated with higher treatment adherence, better quality of life and cognitive functioning [20,31].

The relationship between premorbid adjustment and sociodemographic characteristics such as age, gender and ethnicity has also been analyzed. Childhood and adolescence are the most predictive periods for the development of Schizophrenia [21,23,30,32], where earlier onset of the disorder correlates with worse premorbid adjustment [20,23]. It has been observed that males tend to have worse premorbid adjustment [21,33,34] and that females have better recovery rates, especially in childhood and adolescence [30]. However, there are differences in the detection of this disorder in females, resulting in a longer delay in the initiation of treatment in the latter [35]. In relation to ethnicity, no relevant differences have been found in the literature, but limitations related to samples stand out [21,23,36].

The study of premorbid adjustment has allowed understanding the heterogeneity of schizophrenia in terms of its therapeutic outcomes [21,23]. However, the availability of research relating premorbid adjustment with recovery measures is limited, where the available studies are mainly focused on the population with a first psychotic episode, which do not consider the Latin American population, with restrictive recovery measures, which do not jointly integrate symptom remission, functionality and the patient’s subjective perspective [23,28,30]. The present study aims to analyze the relationship between premorbid adjustment and recovery in a Latin American population diagnosed with schizophrenia by jointly integrating measures associated with symptom remission, functionality and psychological recovery of the patient, using latent class analysis to better understand how different premorbid adjustment profiles are associated with different measures of recovery.

## 2. Materials and Methods

### 2.1. Design and Participants

An observational cross-sectional design was used to evaluate a group of patients diagnosed with schizophrenia in northern Chile during a 3-month period. The sample of participants corresponds to 217 patients with a diagnosis of schizophrenia according to the criteria of the International Classification of Diseases (ICD), 11th version, with stabilized symptomatology, users of three outpatient facilities of the Public Mental Health Service of Arica. A non-probabilistic sampling by availability was used. The mean age was 41.1 years (SD = 16.34) of which 129 patients (56.8%) were men, 181 (79.7%) did not have a partner and 86 (37.9%) self-reported belonging to the Aymara ethnic group. Overall, the age of onset of the first acute psychotic episode was 21.4 years (SD = 8.4) and the age of treatment was 25 (SD = 8.9). All patients received antipsychotics, 65 (30.5%) received psychotherapy and 44 (20.6%) received occupational therapy. Only 1 participant (0.4%) had severe psychotic symptoms, 15 (6.6%) had marked psychotic symptoms, 33 (14.5%) had moderate psychotic symptoms and 176 (77.5%) had mild symptoms. Table 1 provides more information on the characteristics of the sample. A history of organic damage, active drug and alcohol use, and organic-based psychosis were established as exclusion criteria.

### 2.2. Ethical Considerations

The study was approved by the Ethics Committee of the University of Tarapacá (18/2009) and the National Health Service of Chile. Written informed consent was obtained from the patients and their primary caregivers. The objectives of the study were explained, as well as the voluntary nature of participation. No compensation was offered for participation in the study.

### 2.3. Measures

Premorbid Adjustment Scale (PAS) [18]: This scale assesses the degree to which a person has successfully achieved certain developmental milestones at various life stages preceding the initial onset of psychosis symptoms. Thus, functioning is assessed in four age periods or subscales: childhood (up to 11 years), early adolescence (12–15 years), late adolescence (16–18 years) and adulthood (19 years and older), in five major psychosocial domains: sociability and withdrawal, peer relations, school performance, school adjustment and socio-sexual adjustment. Socio-sexual functioning is not included as a psychological domain during the childhood period, just as school performance and school adjustment are not measured during the adulthood period. The PAS scale includes 26 items that have a score range from 0 to 6, where “0” denotes normal adjustment and “6” severe impairment. The rater selects the number that most closely matches the closest descriptive statement. The overall PAS score is calculated by averaging the scores obtained in each of the developmental subscales and in the general section. Higher scores represented lower levels of premorbid adjustment. The scale was adapted to Spanish by Barajas et al. [37]. The scale reported acceptable levels of internal consistency: Total PAS scale (α = 0.89), Sociability and withdrawal (α = 0.89), Peer relationships (α = 0.89), School performance (α = 0.84), School adjustment (α = 0.86) and Sexual adjustment (α = 0.76).

Recovery Assessment Scale (RAS-24) [38,39]: This scale evaluates the subjective assessment of recovery through 24 items that have resulted from the factor analysis of the original scale composed of 41 items. The response options have a 5-level Likert format (1 = “Strongly disagree” to 5 = “Strongly agree”). The total scale of the RAS-24 was used, which presents adequate evidence of reliability and validity [39] and has been translated in Spain by Muñoz et al. [40] and also Zalazar et al. [41] examined the psychometric properties of this instrument in Argentina.

*Positive and Negative Syndrome Scale for Schizophrenia (PANSS)* [42]: This self-report scale was development to assess psychotic symptoms in individuals with schizophrenia. For this purpose, the PANSS scale was used in its five-factor version developed by Lançon et al. [43] which contains the following dimensions: positive symptoms (5 items), negative symptoms (7 items), excitation (5 items), depression (4 items) and cognitive symptoms (3 items). The response options are in 7-level Likert format (1 = “absent” to 7 = “extreme”). The scores to be interpreted are obtained by calculating the sum of all responses. The scores were interpreted according to the cut-off points of Leucht et al. [44], where a PANSS total score of 58 suggests “mildly ill”, a PANSS of 75 to “moderately ill”, a PANSS of 95 to “markedly ill” and a PANSS of 116 to “severely ill”. The PANSS has been translated and validated in Spain by Peralta and Cuesta [45] and also Fresán et al. [46] examined the psychometric properties of this instrument in Mexico.

*Global Activity Assessment Scale (GAF)* [47]: Developed by the American Psychiatric Association (APA), it is an assessment of global functioning on axis V of the diagnosis of patients with severe mental disorders that is typically used for outpatients and inpatients assessing three domains: social functioning, personal functioning and psychological functioning. It presents a single item that assesses global activity and satisfaction in multiple activities it is scored on a scale from 0 to 100, with higher scores reflecting higher function in all domains. Particularly for schizophrenia, it has demonstrated reliability for assessing the level of functioning in these patients [48].

*Stigma resistance:* The stigma resistance subscale of the Internalized Stigma Scale for Mental Illness (ISMI-29) was used [49]. The ISMI is a 29-item self-rated questionnaire comprising five subscales (Alienation, Endorsement of Stereotypes, Experience of Discrimination, Social Withdrawal and Resistance to Stigma). Each item is scored on a four-point Likert scale from 1 (strongly disagree) to 4 (strongly agree). The ISMI has been validated and its internal consistency and retesting are acceptable for the original version [49]. A high total score on the ISMI scale indicates more severe internalized stigmatization. The original version of the ISMI scale was translated into Spanish by Bengochea-Seco et al. [50].

*Duration of Untreated Psychosis (DUP):* The beginning of the DUP period was defined as the first appearance of positive symptoms, and the end was marked by the date of the first hospitalization or the start of antipsychotic treatment [51]. For greater precision, information on the appearance of the first symptoms and the start of treatment, the calculation of the score was obtained by averaging the information provided by the patient, his or her primary caregiver and the clinical record.

*Sociodemographic Covariates:* ethnicity (1 = not belonging to an ethnicity; 2 = belonging to an ethnic group) and biological sex (1 = male; 2 = female).

### 2.4. Procedure

At each center, during a three-month window, all patients were invited to participate when they attended their monthly follow-up medical check-ups. Two psychologists, who were part of the research team and were supervised by the principal investigator, conducted the assessment of the patients at their respective mental health centers, lasting between 40 and 60 min.

Before starting the survey, written informed consent was requested and received from the patient. The objectives of the study and the voluntary nature of participation were explained.

### 2.5. Statistical Analyses

Latent Profile Analysis (LCA) was used to identify premorbid fit profiles in individuals diagnosed with schizophrenia by estimating a model of up to a total of four classes to identify the model with the best fit. LCA can be considered similar to cluster analysis in that both are considered “person-oriented analyses“ [52] that use patterns of case scores to identify individuals who can be grouped together; however, cluster analysis and LCA make different assumptions about the data and use different statistical procedures. In LCA, probabilities of class membership are obtained, not clear assignments as in the case of cluster analysis [52,53]. On the other hand, LCA offers stronger theoretical foundations by providing better defined measures of model fit, the ability to perform confirmatory analyses, and to determine whether a LCA solution is equally applicable to multiple known groups using invariance assessment techniques [54].

A combination of criteria was used to determine the number of best-fit latent classes considering three categories: information theoretical criteria, maximum likelihood statistical methods and entropy-based criteria. The Akaike information criterion (AIC) and the Bayesian information criterion (BIC) are the two original and most widely used information theoretic methods for model selection. Relative fit indicators were considered to optimize decision making on the optimal fit equilibrium of the data set. The BIC rewards parsimony in the models and can be used to compare LCA solutions with competitors. Lower BICs indicate better fit. One can also examine the Akaike information criterion (AIC) and the sample size-adjusted Bayesian information criterion (aBIC) whose lower values also indicate better fit.

The second category of methods for assessing model fit considered the use of statistical likelihood ratio (LR) tests. These tests compare the relative fit of two models that differ by a set of parameter restrictions. The adjusted Lo–Mendell–Rubin (aLMR) test and a bootstrap likelihood ratio test (BLRT) was employed in which a small probability value (*p* < 0.05) indicates that the K-0 class model provides a significantly better fit to the observed data than the K-1 class model.

Thirdly, the entropy diagnostic indicates the accuracy with which the model defines the classes. Generally, a higher value of normalized entropy represents a better fit; values (>0.80) indicate that the latent classes are highly discriminant [55].

In addition to assessing fit, an interesting alternative is to review the classification diagnostics [56]. Although diagnostic statistics are not used to select the final class model, they are important to consider as they estimate class membership for individuals [57]. The average latent posterior probabilities are presented in a matrix with diagonals representing the average probability that a person will be assigned to a class given his or her scores on the criterion variables used to create the classes. Higher diagonal values (i.e., closer to 1.0) are desirable. The off-diagonal elements in the posterior probability matrix contain probabilities about the classified cases being assigned to another class later. Lower off-diagonal values (i.e., closer to 0) are desirable. A cutoff of (0.80) is advised for acceptable diagonal probabilities [55]. Others suggest a cutoff value higher than (0.90) [58]. Although values above (0.90) are ideal, if other criteria are met and the model is theoretically supported, probabilities between (0.80) and (0.90) are considered acceptable.

One-factor ANOVA and chi-square tests will also be used for sample comparisons between latent profiles.

LCA was performed using Mplus software V.2 [59] that allows for the use of both continuous and categorical latent variables using maximum likelihood estimation with robust standard errors as the estimation method [60]. For the analysis of covariables, multinomial logistic regressions were performed for the group of covariables. The three-step method was used to explore the relationships between the latent class variable and the predictor variables. In this approach, the latent class model is estimated in a first step using only the latent class indicator variables. In the second step, the most likely class variable is created using the posterior distribution of the latent class obtained during the first step. In the third step, the most probable class is refitted to the predictor variables taking into account the misclassification from the second step.

## 3. Results

The results of the correlation analysis between the PAS total score and the clinical indicators of recovery showed that there is a significant correlation between most of the study variables and the PAS total score, with the exception of years of untreated psychosis (DUP). Table 2 presents the correlation matrix between the PAS total score and the rest of the study variables.

Table 2 shows the correlations between the total PAS scale score and the rest of the study variables.

The results of the LCA support the convenience of the latent class analysis for the group of people diagnosed with Schizophrenia. Table 3 presents the results of the LCA, in which it can be seen that the three-class model presents the best indicators of relative fit (AIC = 3618.92; aBIC = 3693.28; aLMR = 133.22; BLR = −1856.13).

Figure 1 represents the distribution of the means of the three latent profiles for the premorbid fit measurements. The (X) axis represents the label for each dimension and the (Y) axis provides the means. It can be stated that in (C1) individuals with the most favorable levels of premorbid adjustment were grouped (*n* = 93). In (C2), individuals with moderate levels of premorbid adjustment were grouped (*n* = 100). On the other hand, (C3) grouped individuals with poorer levels of premorbid adjustment (*n* = 24). In general, there are differences between the dimensions assessed by the PAS between the different evolutionary periods of the patients; however, there are some periods in which no significant differences were observed between the latent profiles. For example, with respect to sociability and withdrawal, no differences were observed between (C2) and (C3) during adulthood (HSD= −0.523; *p* > 0.05). Nor were differences in school performance observed between (C1) and (C2) during early adolescence (HSD = −0.218; *p* > 0.05) and during late adolescence (HSD= −0.382; *p* > 0.05). Regarding school adjustment, no differences were observed between (C1) and (C2) during early adolescence (HSD = −0.024; *p* > 0.05) and late adolescence (HSD= −0.387; *p* > 0.05). In relation to psychosexual functioning, no differences were observed between (C2) and (C3) during late adolescence (HSD= −0.413; *p* > 0.05) and adulthood (HSD= −0.045; *p* > 0.05).

Table 4 presents the distribution of means for the total sample and for the classification of latent profiles in each subdimension of the PAS and the study covariates. Statistically significant differences were observed in all dimensions of the PAS among the premorbid adjustment latent profiles. Statistically significant differences were also observed between latent profiles on all clinical variables. No significant differences were observed between the latent profiles of premorbid adjustment with respect to sex or ethnicity.

Table 5 presents the results of the analysis of covariates as predictors of class membership of the pre-morbid adjustment profiles. Patients classified in the best pre-morbid adjustment group (C1) are more likely to have better levels of subjective recovery (β = −0.055; *p* < 0.001) compared to (C2). On the other hand, patients classified in (C3) who have poor pre-morbid adjustment are significantly more likely to have low levels of stigma resistance compared to patients with better pre-morbid adjustment (C1) (β = 0.349; *p* < 0.001) and moderate pre-morbid adjustment (C2) (β = −0.338; *p* < 0.001).

## 4. Discussion

The main hypothesis of the study was that patients with good premorbid adjustment will have better indicators of recovery. The results support this hypothesis since premorbid adjustment, as measured by PAS total score, was found to correlate significantly with subjective recovery, stigma resistance, functionality and symptomatology. However, time in untreated psychosis (DUP) was not significant. Regarding the relationship between premorbid adjustment and disorder severity, the results were convergent with previous studies [20,21]. On the other hand, this study provides new evidence that premorbid adjustment is not only related to objective measures of disorder severity, but also to more subjective measures such as stigma resistance and recovery [16].

The results of the LCA analysis showed that a three-profile classification was adequate to describe latent premorbid adjustment scores, the results showed that the profile with the most favorable premorbid psychosocial functioning (C1) was more likely to have a higher subjective recovery score and greater resistance to stigma compared to the more moderate (C2) or lower functioning profiles (C3). Evidence relating recovery to premorbid adjustment is sparse [61]; similarly, few studies relate premorbid adjustment to stigma [27,62]. These results highlight the importance of premorbid functioning in the development of a favorable subjective outlook on self and recovery from the disorder. In this sense, premorbid social adjustment as a concept is a measure that assesses issues such as how a person interacts with schoolmates, builds meaningful relationships and the presence of age-relevant sexual interest. Previous studies have shown that poor premorbid adjustment and problems maintaining social competence affect individuals’ schematic beliefs about themselves and others. These schematic beliefs have an important effect on self-esteem and self-concept [63], so it is possible that premorbid adjustment as a measure reflects the influence of premorbid social competence on the construction of self-concept, a key component in and resistance to stigma subjective recovery [64].

Moreover, in this work, cognitive impairment assessed by the PANSS correlated significantly with premorbid adjustment [62]. In this sense, premorbid adjustment also assesses school performance and adjustment during childhood and adolescence, being a predictor of cognitive impairment, deficits in social cognition and severity in positive and negative symptomatology [65]. In this sense, premorbid adjustment is a measure that reflects part of the cognitive competence necessary to construct positive future beliefs about the disease, the treatment and a firm belief in the possibility of recovery; on the contrary, those patients with poor premorbid adjustment usually present a marked cognitive impairment that facilitates identification with negative stereotypes and the development of self-stigmatizing schemas. It is therefore consistent that premorbid adjustment is associated with other measures such as subjective recovery and resistance to stigma. However, it would be convenient to deepen these findings by incorporating in future research other variables associated with cognitive functioning such as IQ [66] and drug and alcohol use [67].

The study has limitations that make it difficult to generalize the results to the entire schizophrenia population. The use of retrospective interviews to establish premorbid functioning could be affected by recall biases that would not allow accurate measurement; however, this limitation is mitigated to some extent by studies demonstrating strong correlations between PAS scores and, for example, psychosocial functioning scores obtained before the onset of schizophrenia [68]. Another issue is that the characteristics of the cross-sectional design do not allow for adequate assessment of the stability of premorbid adjustment profiles and their relationship to study covariates. Future research should address a longitudinal approach that would allow us to assess the stability of the profiles and the prediction of covariates using latent transition analysis (LTA).

The implications of this study raise important aspects for clinical practice. On one hand, these findings converge with previous studies that consider premorbid adjustment in schizophrenia as a neuropsychiatric precursor measure of behavior that may predict later manifestations of the disorder [69]. These findings also reinforce the idea that strengthening social adjustment during the prodromal phase may improve patients’ subjective assessment of the disorder [70]. Likewise, the construction of profiles based on premorbid adjustment may contribute to better identify the psychosocial treatment, psychotherapy or cognitive rehabilitation needs of people diagnosed with treatment-resistant schizophrenia [71] or who come from socio-health contexts where the allocation of resources for the treatment of mental disorders is limited, as in the case of Latin America [72].

## 5. Conclusions

Results show that premorbid adjustment was significantly correlated with symptomatology severity, functionality, stigma resilience and subjective recovery. Additionally, it is observed that latent profiles of better premorbid adjustment predict better outcomes in subjective recovery and stigma resilience. The results show that premorbid adjustment not only has implications for the severity of the disorder, but that psychosocial functioning prior to psychosis affects the patient’s subjectivity, the representation of the disorder, and the recovery process. In addition, the latent classification model is suitable for assessing premorbid adjustment profiles to guide future interventions in the schizophrenia population.

## Figures and Tables

**Figure 1 jcm-11-03840-f001:**
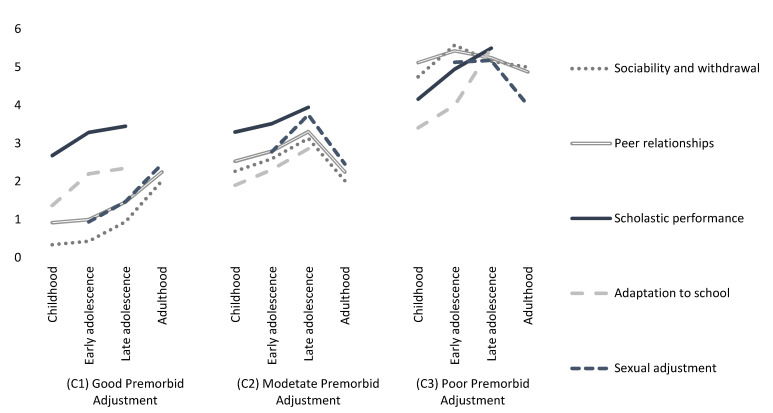
Means of latent profiles for pre-morbid adjustment scores in the sample.

**Table 1 jcm-11-03840-t001:** Sample characteristics.

		*M (s.t)* ± Rank, *n* (%)
Gender	Male	122 (56.2%)
	Female	95 (43.8%)
Ethnicity	Does not identify with any Ethnicity	119 (55.1%)
	Recognizes with any Ethnicity	97 (44.9%)
Categorization of symptomatology	Mild	167 (77.0%)
	Moderate	32 (14.7%)
	Marked	15 (7%)
	Severe	1 (0.5%)

Note: M = Mean; *s.t* = Standard Deviation; *n* = Number of subjects; % = frequency in percent.

**Table 2 jcm-11-03840-t002:** Correlation Matrix.

	PAS TOTAL	
ISMI Stigma resistance	−0.453	***
RAS total scale	−0.394	***
GAF	−0.444	***
DUP	0.191	
PANSS Positive	0.225	*
PANSS Negative	0.244	*
PANSS Cognitive	0.382	***
PANSS Excited component	0.211	*
PANSS Depression	0.430	***

Note. * *p* < 0.05, *** *p* < 0.001.

**Table 3 jcm-11-03840-t003:** Model fit for different solutions of k Stigma profiles.

*K*-Class	LogLikelihood	Entropy	AIC	aBIC	aLMR Test	*p*	BLR Test	*p*
2	−1856.136	0.818	3744.27	3798.35	257.22	0.092	−1988.73	0.000
3	−1787.461	0.854	3618.92	3693.28	133.22	0.000	−1856.13	0.000
4	−1764.192	0.818	3584.38	3590.29	45.14	0.237	−1787.46	0.000

Note. AIC = Akaike Information Criterion; BIC = Bayesian Information Criterion; LMR = Lo–Mendel–Rubin likelihood test; BLR = Bootstrapped likelihood ratio.

**Table 4 jcm-11-03840-t004:** Descriptive statistics of clinical variables.

*Mean (s.d)*/*(%)*					
PAS	Total Sample	C1	C2	C3	F
Sociability and withdrawal	2.16	0.75 (0.11)	2.76 (0.17)	5.09 (0.24)	252.943 **
Peer relationships	2.43	1.25 (0.12)	2.86 (0.13)	5.12 (0.25)	185.617 **
Scholastic performance	3.48	3.06 (0.18)	3.57 (0.17)	4.71 (0.22)	12.175 **
Adaptation to school	2.32	1.86 (0.16)	2.34 (0.15)	3.98 (0.30)	22.340 **
Sexual adjustment	2.73	1.45 (0.12)	3.30 (0.17)	4.92 (0.30)	91.772 **
Covariate	Total Sample	C1	C2	C3	F
SR	12.77 (2.90)	13.70 (2.4)	12.59 (2.81)	9.82 (3.0)	19.349 **
DUP	2.45 (6.30)	1.52 (3.18)	2.33 (4.8)	3.35 (8.42)	2.033
RAS	76.71 (15.97)	83.3 (14.44)	73.27 (15.32)	64.70 (14.93)	19.771 **
GAF	67.26 (14.71)	70.10 (15.07)	67 (13.66)	55 (13.18)	10.735 **
PANSS	61.80 (18.91)	57.10 (17.83)	63.90 (18.55)	75.20 (18.41)	10.082 **
Covariate	Total Sample	C1	C2	C3	X^2^
Male%	122 (56.2%)	58 (26.7%)	54 (24.9%)	10 (4.6%)	0.158
Ethnicity%	97 (44.9%)	39 (18.1%)	47 (21.8%)	11 (5.1%)	0.604

Note: ** *p* < 0.01. PAS = Premorbid adjustment total scale; SR = Stigma resistance; RAS = Recovery assessment total scale; DUP = Duration of Untreatment Psychosis; GAF = Global Activity Assessment Scale; PANSS = Positive and Negative Syndrome Scale for Schizophrenia.

**Table 5 jcm-11-03840-t005:** Results of covariate analysis as predictors of latent classification.

Reference Group	Comparison Group	RAS	PANSS	GAF	RS	Ethnicity	Gender
C1	C2	−0.055 *	0.029	0.024	−0.029	0.253	0.397
	C3	−0.037	0.030	−0.013	−0.349 *	0.152	0.879
C3	C2	−0.019	−0.001	0.037	0.338 *	0.101	−0.482

Note: * *p* < 0.05.

## Data Availability

Not applicable.

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
