# Peer review of "Latent Profiles of Premorbid Adjustment in Schizophrenia and Their Correlation with Measures of Recovery"

_jcm, 2022, doi:10.3390/jcm11133840_

Round 1

Reviewer 1 Report

The paper addresses a basically familiar topic. Student textbooks describe the effect of good premorbid functioning on better prognosis in schizophrenia. However, the authors demonstrate in a methodologically correct way that this topic can still be explored. The results obtained are as expected and the way they are presented is not objectionable. 

The only comment on the work is the lack of information about the accuracy and reliability of PAS. 

Reviewer 2 Report

In this paper the authors want to correlate latent profiles of premorbid adjustment to recovery in schizophrenia patients. The idea is very interesting. 

Here are some tips:

1- In my opinion, in the text, there are some statements that are not clear.

Please explain better the phrase:

"While schizophrenia often develops with no apparent indicator of the impending disorder, premorbid abnormalities are  often present in apparently healthy adolescents who later develop schizophrenia, so premorbid adjustment in schizophrenia is considered a psychosocial or phenotypic expression of brain alteration that precede(s) psychosis"

2- In the Introduction section, a paragraph on what recovery is should be included.

3- The authors should insert references for DUP when defining DUP ("Years of Untreated Psychosis"). Moreover, I could not find in the text and in table 4 the years of DUP.

4- In my opinion it is fundamental to make the difference between DUP and duration of untreated illness.

5- It is not clear the duration of the study. Please explain that this is an observational study

Kind regards.

Reviewer 3 Report

Dear Authors,

Thank you for your contribution to the field. I found this to be a well-designed study, with sound methodology, which was easy to read and well presented. I noticed minor spelling mistakes which I have highlighted in red in the attached file, and I also included suggestions to strengthen the paper which are more content-related. Specifically, I was missing in the background and discussion mentioning of the role of premorbid IQ, and also alcohol and drug use in adulthood which has also been found to have a strong impact on functioning in people with schizophrenia.

Author Response

In attachment.

This manuscript is a resubmission of an earlier submission. The following is a list of the peer review reports and author responses from that submission.

Round 1

Reviewer 1 Report

This manuscript adds nicely to the evidence that premorbid  adjustment correlates with outcome parameters in schizophrenia using sophisticated statistical methods. 

Major points:

My major point of criticism is the title of the manuscript. In the discussion section the authors admit as a limitation that the cross-sectional design of their study with retrospective  assessment of premorbid functioning limits its meaningfulness. To this end it's not correct to speak of predictors in the title. Rather, the authors should call it correlation, e.g. "Latent profiles of premorbid adjustment in Schizophrenia are correlated with recovery measures"

The second section of the Introduction is a pure repetition of what is written in the abstract. This should be improved - as the introduction is rather short it could deserve more detail, especially the definition of premorbid adjustment in lines 36-38. It stays unclear to the reader what the authors mean by "psychosocial or phenotypic expression of brain alterations that precede psychosis". 

At the end of the introduction a short section explaining the aims and novelty of this study (what is different to comparable studies?) would be helpful for the reader.

minor points:

Abstract line 12: correct to: "it is a concept which possesses..."

Introduction line 30: delete "their"

Introduction line 40: replace "worsening" by "worse"

introduction lines 41-42: should be: "severity in theory of mind deficits"

methods lines 76-77: replace "non-inclusion criteria" by "exclusion criteria"

methods line 113: replace "development" by "developed"

Author Response

Major points:

  1. My major point of criticism is the title of the manuscript. In the discussion section the authors admit as a limitation that the cross-sectional design of their study with retrospective  assessment of premorbid functioning limits its meaningfulness. To this end it's not correct to speak of predictors in the title. Rather, the authors should call it correlation, e.g. "Latent profiles of premorbid adjustment in Schizophrenia are correlated with recovery measures"

R: Many thanks to the reviewer for the suggestion, we have made a modification to the title of the article.

  1. The second section of the Introduction is a pure repetition of what is written in the abstract. This should be improved - as the introduction is rather short it could deserve more detail, especially the definition of premorbid adjustment in lines 36-38. It stays unclear to the reader what the authors mean by "psychosocial or phenotypic expression of brain alterations that precede psychosis". 

R: Taking into account the reviewer's correction, the wording of the reference has been modified as follows:

"While schizophrenia often develops with no apparent indicator of the impending disorder, premorbid abnormalities are often present in apparently healthy adolescents who later develop schizophrenia, so premorbid adjustment in schizophrenia is considered a psychosocial or phenotypic expression of brain alterations that precede psychosis"

  1. At the end of the introduction a short section explaining the aims and novelty of this study (what is different to comparable studies?) would be helpful for the reader.

R: Considering what the reviewer pointed out, the details of the research objective have been reformulated, emphasizing how this study uses broad measures of recovery, better balances the heterogeneity of the schizophrenia population using LCA and encompassing an understudied population such as Latin America.

“The study of premorbid adjustment has allowed understanding the heterogeneity of schizophrenia in terms of its therapeutic outcomes [17, 19], however the availability of research relating premorbid adjustment with recovery measures is limited, where the available studies are mainly focused on the population with a first psychotic episode, which do not consider the Latin American population, with restrictive recovery measures, which do not jointly integrate symptom remission, functionality and the patient's subjective perspective [24, 26, 29]. The present study aims to analyze the relationship between premorbid adjustment and recovery in a Latin American population diagnosed with schizophrenia by jointly integrating measures associated with symptom remission, functionality and psychological recovery of the patient, using latent class analysis to better understand how different premorbid adjustment profiles are associated with different measures of recovery.”

minor points:

Abstract line 12: correct to: "it is a concept which possesses..." (Was corrected)

Introduction line 30: delete "their" " (Was corrected)

Introduction line 40: replace "worsening" by "worse" (Was corrected)

introduction lines 41-42: should be: "severity in theory of mind deficits" (Was corrected)

methods lines 76-77: replace "non-inclusion criteria" by "exclusion criteria" (Was corrected)

methods line 113: replace "developed" by "developed" (Was corrected)

Reviewer 2 Report

There have been so many studies with the similar contents and aims that the authors need to summarize them more clearly in the introduction. Since they have done a retrospective analysis, it is considered to be less reliable. Also, it is unclear which of their results show originality and importance.

Author Response

There have been so many studies with the similar contents and aims that the authors need to summarize them more clearly in the introduction. Since they have done a retrospective analysis, it is considered to be less reliable. Also, it is unclear which of their results show originality and importance.

RESPONSE

We are grateful for the reviewer's critical opinion. We believe that recovery, integrating measures of remission, functionality and subjective recovery, is not adequately addressed in the premorbid adjustment of schizophrenia in the Latin American context. We have modified the wording of the objective by emphasizing that the schizophrenia population, being very heterogeneous, the latent class analysis approach may better contribute to the discussion.